# Towards Tight Communication Lower Bounds for Distributed Optimisation

**Dan Alistarh**
IST Austria & Neural Magic, Inc.
dan.alistarh@ist.ac.at

**Janne H. Korhonen**
IST Austria
janne.korhonen@ist.ac.at

## Abstract

We consider a standard distributed optimisation setting where $N$ machines, each holding a $d$-dimensional function $f_i$, aim to jointly minimise the sum of the functions $\sum_{i=1}^{N} f_i(x)$. This problem arises naturally in large-scale distributed optimisation, where a standard solution is to apply variants of (stochastic) gradient descent. We focus on the *communication complexity* of this problem: our main result provides the first fully unconditional bounds on total number of bits which need to be sent and received by the $N$ machines to solve this problem under point-to-point communication, within a given error-tolerance. Specifically, we show that $\Omega(Nd \log d/N\varepsilon)$ total bits need to be communicated between the machines to find an additive $\epsilon$-approximation to the minimum of $\sum_{i=1}^{N} f_i(x)$. The result holds for both deterministic and randomised algorithms, and, importantly, requires no assumptions on the algorithm structure. The lower bound is tight under certain restrictions on parameter values, and is matched within constant factors for quadratic objectives by a new variant of quantised gradient descent, which we describe and analyse. Our results bring over tools from communication complexity to distributed optimisation, which has potential for further applications.

## 1 Introduction

The ability to efficiently distribute large-scale optimisation over several computing nodes has been one of the key enablers of recent progress in machine learning, and the last decade has seen significant attention dedicated to efficient distributed optimisation. One specific area of focus has been on reducing the *communication cost* of distributed machine learning, i.e. the total number of bits sent and received by machines in order to jointly optimise an objective function. To this end, communication-efficient variants are known for most classical optimisation algorithms, and in fact entire *families* of communication-compression methods have been introduced in the last decade.

We consider a standard setting in which $N$ machines communicate by sending point-to-point binary messages to each other. Given dimension $d$, and a domain $\mathbb{D} \subseteq \mathbb{R}^d$, each machine $i$ is given an input function $f_i \colon \mathbb{D} \to \mathbb{R}$, corresponding to a subset of the data, and the machines need to jointly minimise the empirical risk $\sum_{i=1}^{N} f_i(x)$ with either deterministic or probabilistic guarantees on the output, within $\varepsilon$ additive error tolerance. That is, *at least one node* needs to output $z \in [0,1]^d$ such that

$$\sum_{i=1}^{N} f_i(z) \le \inf_{x \in [0,1]^d} \sum_{i=1}^{N} f_i(x) + \varepsilon \,. \tag{1}$$

This setting models data-parallel optimisation, and covers virtually all practical settings, from large-scale regression, to the training of deep neural networks.

35th Conference on Neural Information Processing Systems (NeurIPS 2021).

The key parameters governing communication complexity are the problem dimension $d$, the solution accuracy $\varepsilon$, and the number of machines $N$. Most communication-efficient approaches can be linked to (at least) one of the following strategies: *dimensionality-reduction* methods, such as the *sparsification* of model updates [3, 14, 15, 17, 18], or *projection* [10, 34, 35], which attempt to reduce the dependency on the parameter $d$, *quantisation* methods [2, 11, 27, 31], whose rough goal is to improve the dependency on the accuracy $\varepsilon$, and *communication-reduction methods* such as reducing the *frequency of communication* [5, 30, 36] relative to the number of optimisation steps, or communicating via point-to-point messages via, e.g., *gossiping* [19, 24].

Although these methods use a diverse range of algorithmic ideas, upon close inspection, they all appear to have a worst-case total communication cost of at least $Nd\log(d/\varepsilon)$ bits, even for simple convex $d$-dimensional problems, and even for variations of the above standard setting. For instance, even though dimensionality-reduction or quantisation methods might send *asymptotically less* than $d$ bits per algorithm iteration, they have to compensate for this in the worst case by running for asymptotically more iterations. (See e.g. [2] for a simple example of this trade-off.) It is therefore natural to ask whether this complexity threshold is inherent, or whether it can be circumvented via improved algorithmic techniques. This is our motivating question.

A partial answer is given by the foundational work of Tsitsiklis and Luo [32], who gave a lower bound of $\Omega(d\log(d/\varepsilon))$ in the case where *two nodes* communicate to optimise over quadratic functions. Their argument works by counting the total number of possible $\varepsilon$-approximate solutions in the input volume: communication has to be at least the logarithm of this number. Subsequent work has considered more complex input functions, e.g. [41], or stronger notions of approximation [33]. The original argument generalises directly to $N$ nodes under the strong assumption that *each node has to return the correct output*: in this case, communication complexity is asymptotically $N$ times the 2-node cost [11].

In this context, it is surprisingly still unknown whether the $\Omega(Nd\log(d/\varepsilon))$ communication threshold is actually inherent for distributed optimisation in the standard case where only a single node needs to return the output. This question is not just of theoretical interest, since there are many practical settings in large-scale optimisation, such as *federated learning* [20] or the *parameter server* setting [23], where only a single node coordinates the optimisation, and knows the final answer. This raises the question whether more communication-efficient algorithms are possible in such settings, in which communication cost is a key concern. At the same time, it is also not clear under which conditions algorithms achieving this asymptotic complexity exist.

## 1.1 Contribution

In this paper, we take a significant step towards addressing these questions. Our main result is the first *unconditional* lower bound on the communication complexity of distributed optimisation in the setting discussed above, showing that it is impossible to obtain a significant improvement in communication cost even if only one node (the "coordinator") learns the final output. Specifically, even if the input functions $f_i$ at the nodes are promised to be quadratic functions $x \mapsto \beta_0\|x - x^*\|_2^2$ for some constant $\beta_0 > 0$, then any deterministic or randomised algorithm where at least one node learns a solution to (1) requires

$$\Omega\Big(Nd\log\frac{\beta d}{N\varepsilon}\Big) \text{ total bits to be communicated}$$

for $\beta = \beta_0 N$, as long as parameters satisfy $\beta d/N^2\varepsilon = \Omega(1)$. We emphasise that the lower bound requires *no* assumptions on the structure of the algorithm or amount of local computation. We also note that in most practical settings, the parameter dependency requirement is satisfied, as the number of parameters $d$ is significantly larger than the number of machines $N$ multiplied by the error tolerance $\varepsilon$ – moreover, a non-trivial dependence between $\beta$, $d$, $N$ and $\varepsilon$ is required for the lower bound to hold. We discuss this in detail below.

Our results start from the classic idea of linking communication complexity with the number of quadratic functions with distinct minima in the domain [32]. To extend this approach to *randomised (stochastic)* algorithms and to the multi-node case $N > 2$, we build new connections to results and techniques from *communication complexity* [21]. Such connections have not to our knowledge been explored in the context of (real-valued) optimisation tasks. Our work thus provides a template and a basic toolkit for applying communication complexity results to distributed optimisation. As

further applications, we improve the main lower bound to $\Omega(Nd\log(\beta d/\varepsilon))$ for the deterministic case if some node is required to output both the approximate minimiser $z$ and the value of the sum $\sum_{i=1}^{N} f_i(z)$, as well as prove stronger lower bounds in the more challenging *non-convex* case (see Section 5 and Appendices A and B.)

To complement this lower bound, we show that for strongly convex and strongly smooth functions, distributed optimisation can be done using deterministic quantised gradient descent with

$$O\Big(Nd\kappa \log \kappa \log \frac{\beta d}{\varepsilon}\Big) \text{ total bits communicated,}$$

where $\sum_{i=1}^{N} f_i$ is $\alpha$-strongly convex and $\beta$-strongly smooth, and $\kappa = \beta/\alpha$ is the condition number. This is, to our knowledge, the first tight upper bound for communication cost of quantised gradient descent on quadratic functions, as well as the first upper bound that does not require all-to-all broadcast. In particular, for constant condition number $\kappa$, this matches our main lower bound when $d \gg N$, e.g. $d = \Omega(N^{2+\delta})$ for constant $\delta > 0$.

Our algorithm builds on prior quantised gradient descent implementations [2, 25], however, to achieve a tight bound, we need to (a) ensure that our gradient quantisation is sufficiently parsimonious, using $O(d \log \kappa)$ bits per gradient, and (b) avoid all-to-all exchange of gradients. For (a), we specialise a recent lattice-based quantisation scheme which allows arbitrary centring of iterates [11], and for (b), we use two-stage quantisation approach, where the nodes first send their quantised gradients to the coordinator, and the coordinator then broadcasts the carefully quantised sum back to nodes. In Appendix D, we further show that this running time can be improved using randomisation when $\beta d/N\varepsilon$ is small, using a simple sub-sampling approach.

## 1.2 Discussion

**Implications.** While the focus of our work is theoretical, our results show that current *practical* algorithmic approaches are already close to worst-case optimal. Specifically, we show that it is impossible to obtain algorithms with communication cost e.g. $O(Nd + d \log d/\varepsilon)$ by "pipelining" communication costs across algorithm iterations, performing additional local optimisation steps, or by introducing entirely new algorithmic techniques.

At the same time, our upper bound conceptually shows that, for quadratic functions, carefully quantised gradient descent can cost asymptotically the same as broadcasting the solution, while the lower bound shows that this cost is inherent. Specifically, broadcasting a single $d$-dimensional point from $[0, 1]^d$ within accuracy $(\varepsilon/\beta)^{1/2}$, as required for $\varepsilon$-approximation of the sum $\sum_{i=1}^{N} f_i(z)$, to $N$ nodes costs $\Omega(Nd \log(\beta d/\varepsilon))$ bits [11], and thus the communication cost of our algorithm is tight. The lower bound shows that little can be gained by avoiding this broadcast.

**Extensions.** Following Tsitsiklis and Luo [32] and Magnússon et al. [25], we have assumed above that the range of the input functions is $[0, 1]^d$, and the global objective is the sum of input functions. However, the results apply even with some modifications to the setting.

First, we can consider a case where the global objective is the average $1/N \sum_{i=1}^{N} f_i$ instead of sum. In this case, the lower bound holds with $\beta = \beta_0$, and the upper bound holds as stated, with $\beta$ being the smoothness parameter of the average. Second, the results can be extended for any compact convex domain of input functions, with the precise bounds depending on the *volume* and *diameter* of the domain for the lower and upper bound, respectively. For example, one can easily verify that, for inputs defined over the unit sphere, the bounds still hold, but without the factor $d$ inside the logarithm.

**Limitations.** We note that there still remains a small gap between our upper and lower bounds. To illustrate this, consider optimisation of the *average* of quadratic functions $x \mapsto \|x - x^*\|_2^2$ defined over the unit hypercube $[0, 1]^d$; in this case, the bounds take the form

$$\Omega\Big(Nd \log \frac{d}{N\varepsilon}\Big) \qquad \text{and} \qquad O\Big(Nd \log \frac{d}{\varepsilon}\Big).$$

Moreover, the lower bound requires the parameter dependency $\beta d/N^2\varepsilon = \Omega(1)$ to hold. We note that we cannot fully get rid of this requirement: specifically, as we show in Appendix D, we can use a simple input sub-sampling approach to show that, if we have $\beta d/\varepsilon = O(N^\delta)$ for $\delta < 1$, then problem (1) can be solved with $O(N^\delta d \log \beta d/\varepsilon)$ total bits communicated using a randomised algorithm, asymptotically less than dictated by the general lower bound.

Table 1: Comparison of existing upper and lower bounds on total communication required to solve (1). The label 'BC' denotes results for broadcast model, where each sent message is seen by all nodes, and 'MP' denotes results for message-passing model, where only the recipient of the message sees it.

|  |  | Output | Model | Guarantee | Reference |
|---|---|---|---|---|---|
| Lower bound, quadratic inputs | $\Omega(d \log \frac{\beta d}{\varepsilon})$ | all nodes | 2-node | Det. | [32] |
|  | $\Omega(d \log \frac{\beta d}{\varepsilon})$ | all nodes | BC | Rand. | [13] |
|  | $\Omega(Nd \log \frac{\beta d}{\varepsilon})$ | all nodes | MP | Rand. | [11] |
|  | $\Omega(Nd \log \frac{\beta d}{N\varepsilon})$ | **single node** | **MP** | **Rand.** | **this work**, §4 |
| Upper bound, constant $\kappa$ | $O(Nd \log \frac{\beta d}{\varepsilon})$ | all nodes | BC | Rand. | [2, 22] |
| Upper bound, general inputs | $O(\kappa d \log(\kappa d) \log \frac{\beta d}{\varepsilon})$ | all nodes | 2-node | Det. | [32] |
|  | $O(N\kappa d \log(\kappa d) \log \frac{\beta d}{\varepsilon})$ | all nodes | BC | Det | [25] |
|  | $O(Nd\kappa \log \kappa \log \frac{\beta d}{\varepsilon})$ | **all nodes** | **MP** | **Det.** | **this work**, §6 |

A second question our current techniques do not address is the precise dependency on the condition number $\kappa$. Our lower bound techniques do no benefit from large $\kappa$, so new ideas would be required to address this question.

On the upper bound side, the linear dependency on $\kappa$ appears to be inherent for our quantised gradient descent algorithm. However, very recent progress on *quantised second-order methods* [1, 15] shows that it is possible to improve this dependency in general, by leveraging second-order information together with quantisation. Specifically, Alimisis et al. [1], Islamov et al. [15] provide complex quantised variants of Newton-type algorithms, which can achieve linear-in-$d$ communication cost per iteration, under certain assumptions. Thus, these algorithms can asymptotically reach the optimal $Nd \log(d/\epsilon)$ complexity threshold implied by our lower bounds, within logarithmic factors in $\kappa$ and other terms, for a wider range of inputs. This comes at the relative cost of a more complex algorithm, and significant additional local computation.

## 2 Related work

**Optimisation lower bounds.** The first communication lower bounds for a variant of (1) were given in the seminal work of Tsitsiklis and Luo [32], who study optimising sums of convex functions in a two-machine setting. For *deterministic* algorithms, they prove that $\Omega(d \log(\beta_0 d/\varepsilon))$ bits are necessary. Extensions are given by Zhang et al. [41] and Davies et al. [11]. Please see Table 1.

The basic intuition behind these lower bounds is that a node without information about the input needs to receive $\Omega(d \log(\beta_0 d/\varepsilon))$ bits, as otherwise the node cannot produce sufficiently many different output distributions to cover all possible locations of the minimum (cf. Lemma 2.) It is worth emphasising that their bound is on the *received* bits of the *output node*, and does not directly imply anything for other nodes; for example, an algorithm where each node transmits $O\big((d \log(\beta_0 d/\varepsilon))/N\big)$ bits is not ruled out by these previous results. Generalising their approach to match our results seems challenging, as we would have to (a) explicitly require that all nodes output the solution, and (b) ensure that *no node* can use their local input as a source of extra information.

The recent work of Vempala et al. [33] focuses on the communication complexity of solving linear systems, linear regression and related problems. The results are based on communication complexity arguments, similarly to our lower bound. The main technical differences are that (a) linear regression instances over bounded integer weight matrices have a natural binary encoding, and (b) importantly, the approximation ratio for linear regression is defined *multiplicatively*, not *additively*; the main consequence is that the hard instance in their case is the exact solution of the linear system.

**Statistical estimation lower bounds.** In *statistical estimation*, nodes receive random samples from some input distribution, and must infer properties of the input distribution, e.g. its mean. Specifically, for mean estimation, there are *statistical* limits on how good an estimate one can obtain from limited number of samples, although inputs are drawn from a distribution instead of adversarially. Concretely, the results of Shamir [29] and Suresh et al. [31] apply only to restricted types of protocols. Garg

et al. [13] and Braverman et al. [7] give lower bounds for Gaussian mean estimation, where each node receives $s$ samples from a $d$-dimensional Gaussian distribution with variance $\sigma^2$. The latter reference shows that to achieve the minimax rate $\sigma^2 d/Ns$ on mean squared error requires $\Omega(Nd)$ total communication. These results do not imply optimal lower bounds for our setting.

**Lower bounds on round and oracle complexity.** Beyond bit complexity, one previous setting assumes that nodes can transmit vectors of real numbers, while restricting the types of computation allowed for the nodes. This is useful to establish bounds for the number of iterations required for convergence of distributed optimisation algorithms [4, 28], but does not address the communication cost of a single iteration. A second related but different setting assumes the nodes can access their local functions only via specific oracle queries, such as *gradient or proximal queries*, and bound the number of such queries required to solve an optimisation problem [38, 39].

**Upper bounds.** There has been a tremendous amount of work recently on communication-efficient optimisation algorithms in the distributed setting. Due to space constraints, we focus on a small selection of closely-related work. One critical difference relative to practical references, e.g. [2], is that they usually assume gradients are provided as 32-bit inputs, and focus on reducing the amount of communication *by constant factors*, which is reasonable in practice. One exception is [31], who present a series of quantisation methods for mean estimation on real-valued input vectors. Recently, [11] studied the same problem, focusing on replacing the dependence on input norm with a *variance* dependence. We adapt their quantisation scheme for our upper bound.

Tsitsiklis and Luo [32] gave a deterministic upper bound in a *two-node* setting, with $O\big(\kappa d \log(\kappa d) \log(\beta d/\varepsilon)\big)$ total communication cost. Recently, Magnússon et al. [25] extended this to $N$-node case in the *broadcast* model, with $O\big(N\kappa d \log(\kappa d) \log(\beta d/\varepsilon)\big)$ total communication cost. For randomised algorithms and constant condition number, better upper bound of $O(Nd \log(\beta d/\varepsilon))$ total communication cost in the broadcast model follows by using QSGD stochastic quantisation [2] plugged into stochastic variance-reduced gradient descent (SVRG) [16]. See Künstner [22] for a detailed treatment. While these algorithms are for the broadcast model, they can likely be implemented in the message-passing model without overhead by using two-stage quantisation; however, our algorithm also obtains optimal dependence on $d$ *deterministically*.

## 3   Preliminaries and background

**Coordinator model.** For technical convenience, we work in the classic *coordinator model* [6, 12, 26], equivalent to the message-passing setting. In this model, we have $N$ *nodes* as well as a separate *coordinator* node. The task is to compute the value of a function $\Gamma \colon B^N \to A$, where $B$ and $A$ are arbitrary input and output domains; each node $i = 1, 2, \ldots, N$ receives an input $b_i \in B$. There is a communication channel between each of the nodes and the coordinator, and nodes can communicate with the coordinator by exchanging binary messages. The coordinator has to output the value $\Gamma(b_1, b_2, \ldots, b_N)$. Furthermore, all nodes, including the coordinator, have access to a stream of private random bits.

More precisely, we assume without loss of generality that computation is performed as follows:

(1) Initially, each node $i = 1, 2, \ldots, N$ receives the input $b_i$. The coordinator and nodes $i = 1, 2, \ldots, N$ receive independent and uniformly random binary strings $r, r_i \in \{0, 1\}^c$, respectively, where $c$ is a constant.

(2) The computation then proceeds in sequential rounds, where in each round, (a) the coordinator first takes action by either outputting an answer, or sending a message to a single node $i$, and (b) the node $i$ that received a message from the coordinator responds by sending a a message to the coordinator.

A *transcript* $\tau$ for a node is a list of the messages it has sent and received. A *protocol* $\Pi$ is a mapping giving the actions of the coordinator and the nodes; for the coordinator, the next action is a function of its transcript so far and the private random bits $r$, and for node $i$, the next action is a function of its input $b_i$, its transcript so far and the private random bits $r_i$. The protocol $\Pi$ also determines the number of random bits the nodes receive.

We say that a protocol $\Pi$ computes $\Gamma \colon B^N \to A$ with error $p$ if for all $(b_1, b_2, \ldots, b_N) \in B^N$, the output of $\Pi$ is $\Gamma(b_1, b_2, \ldots, b_N)$ with probability at least $1 - p$. The *communication complexity* of a

protocol $\Pi$ is the maximum number of total bits transmitted by all nodes, i.e. the total length of the transcripts, on any input $(b_1, b_2, \ldots, b_N) \in B^N$ and any private random bits of the nodes.

While the model definition may appear restrictive, the protocol restrictions do not matter when the complexity measure is the total number of bits exchanged. Parallel computation can be sequentialised, and direct messages between non-coordinator nodes can be relayed via the coordinator, with at most constant factor overhead. Furthermore, note that the model is *nonuniform*, i.e. each protocol is defined only for specific functions $\Gamma \colon B^N \to A$ and specific input and output sets $B$ and $A$. Any uniform algorithm working for a range of parameters induces a series of nonuniform protocols, regardless of the computational cost, so lower bounds for coordinator model translate to uniform algorithms.

**Communication complexity.** We now recall some basic definitions and results from communication complexity. In the following, we assume that sets $B$ and $A$ are finite, as this is the standard setting of communication complexity.

For a function $\Gamma \colon B^N \to A$, the *deterministic communication complexity* $\mathsf{CC}(\Gamma)$ is the minimum communication complexity of a deterministic protocol computing $\Gamma$. Likewise, the *$\delta$-error randomised communication complexity* $\mathsf{RCC}^\delta(\Gamma)$ is the minimum communication complexity of a protocol that computes $\Gamma$ with error probability $\delta$.

For a distribution $\mu$ over $B^N$, we define the *$\delta$-error $\mu$-distributional communication complexity of* $\Gamma$, denoted by $\mathsf{D}^\mu_\delta(\Gamma)$, as the minimum communication complexity of a deterministic protocol that computes $\Gamma$ with error probability $\delta$ when the input is drawn from $\mu$. Similarly, the *$\delta$-error $\mu$-distributional expected communication complexity of* $\Gamma$, denoted by $\mathsf{ED}^\delta_\mu(\Gamma)$, is the minimum expected communication cost of a protocol that computes $\Gamma$ with error probability $\delta$, where the expectation is taken over input drawn from $\mu$ and the random bits of the protocol.

Yao's Lemma [40] relates the distributional communication complexity to the randomised communication complexity; see Woodruff and Zhang [37] for a proof in the coordinator model.

**Lemma 1** (Yao's Lemma). *For function $\Gamma$ and $\delta > 0$, we have $\mathsf{RCC}^\delta(\Gamma) \geq \max_\mu \mathsf{D}^\mu_\delta(\Gamma)$.*

**Properties of convex functions.** Recall that a continuously differentiable function $f$ is *$\beta$-(strongly) smooth* if $\|\nabla f(x) - \nabla f(y)\|_2 \leq \beta \|x - y\|_2$, and *$\alpha$-strongly convex* if $\left(\nabla f(x) - \nabla f(y)\right)^T (x - y) \geq \alpha \|x - y\|_2^2$, for all $x$ and $y$ in the domain of $f$. For $\alpha$-strongly convex and $\beta$-strongly smooth function $f$, we say that $f$ has *condition number* $\kappa = \beta/\alpha$. If $f_1$ is $\alpha_1$-strongly convex and $\beta_1$-strongly smooth and $f_2$ is $\alpha_2$-strongly convex and $\beta_2$-strongly smooth, then $f_1 + f_2$ is $(\alpha_1 + \alpha_2)$-strongly convex and $(\beta_1 + \beta_2)$-strongly smooth.

A quadratic function $f(x) = \beta \|x - y\|_2^2 + C$ is $\beta$-strongly convex and $\beta$-strongly smooth. For $\varepsilon > 0$, if $f(x) \leq \varepsilon$, then $\|x - x^*\|_2 \leq (\varepsilon/\beta)^{1/2}$. A sum of quadratics $F(x) = \sum_{j=1}^k a_j \|x - y_j\|_2^2$, where $y_j \in \mathbb{R}^d$ and $a_j \geq 0$ for $j = 1, 2, \ldots, k$, is a quadratic function $F(x) = A\|x - x^*\|_2^2 + C$, where $C$ is a constant and $x^* = \sum_{j=1}^k a_j y_j / A$ is the minimum of $F$.

**Point packing.** We will make use of the following elementary result, which bounds the number of points we can pack into $[0, 1]^d$ while maintaining a minimum distance between all points.

**Lemma 2** (Tsitsiklis and Luo [32]). *For $\delta > 0$ and $d \geq 1$, there is a set of points $S \subseteq [0, 1]^d$ such that (1) $\|x - y\|_2 > \delta$ for all distinct $x, y \in S$, and (2) $|S| \geq (d^{1/2}/C\delta)^d$, where $C = (\pi e/2)^{1/2}$ is a constant.*

## 4 Main lower bound

We now prove our main result, by giving a lower bound for communication complexity of any algorithm solving (1) that holds for both deterministic and randomised protocols.

**Theorem 3.** *Given parameters $N$, $d$, $\varepsilon$, $\beta_0$ and $\beta = \beta_0 N$ satisfying $d\beta/N^2\varepsilon = \Omega(1)$, any protocol solving (1) for quadratic input functions $x \mapsto \beta_0 \|x - x_0\|_2^2$ has communication complexity $\Omega\big(Nd\log(\beta d/N\varepsilon)\big)$.*

To formally apply communication complexity tools, we will prove a lower bound for a *discretised* version of (1) – where both the input and output sets are finite – which will imply Theorem 3. Let $N$,

$d$, $\varepsilon$, and $\beta$ be fixed, assume $d\beta/N^2\varepsilon = \Omega(1)$. Furthermore, let $S$ be the set given by Lemma 2 with $\delta = 3N(\varepsilon/\beta)^{1/2}$, and let $T \subseteq [0,1]^d$ be an arbitrary finite set of points such that for any $x \in [0,1]^d$, there is a point $t \in T$ with $\|x - t\| \le (\varepsilon/4\beta)^{1/2}$. By assumption $d\beta/N^2\varepsilon = \Omega(1)$, the set $S$ has size at least 2. Let $D = \lceil \log|S| \rceil = \Theta(d\log(\beta d/N\varepsilon))$. Again, for convenience, assume $2^D = |S|$, and identify each binary string $b \in \{0,1\}^D$ with an element $\tau(b) \in S$.

**Definition 4.** *Given parameters $N, d, \varepsilon, \beta$, we define the problem* $\mathsf{MEAN}_{d,N}^{\varepsilon,\beta}$ *as follows:*

- *The node inputs are from $\{0,1\}^D$, and*

- *Valid outputs for input $(b_1, b_2, \ldots, b_N)$ are points $t \in T$ that satisfy the condition $\|x^* - t\|_2 \le (\varepsilon/\beta)^{1/2}$, where $x^* = \sum_{i=1}^N \tau(b_i)/N$ is the average over inputs.*

First, we observe that any algorithm for solving (1) can be used to solve $\mathsf{MEAN}_{d,N}^{\varepsilon,\beta}$.

**Lemma 5.** *For fixed $N$, $d$, $\varepsilon$, $\beta_0$ and $\beta = \beta_0 N$, any randomised protocol solving (1) for quadratic functions $x \mapsto \beta_0\|x - x_0\|_2^2$ with error probability $1/3$ has communication complexity at least* $\mathsf{RCC}^{1/3}(\mathsf{MEAN}_{d,N}^{\varepsilon,\beta/4})$.

*Proof.* Let $\Pi$ be protocol solving (1) with communication complexity $C$ and error probability $1/3$. We show that we can use it to solve $\mathsf{MEAN}_{d,N}^{\varepsilon,\beta/4}$ with total communication cost $C$ and error probability $1/3$, implying the claim. Given input $(b_1, b_2, \ldots, b_N)$ for $\mathsf{MEAN}_{d,N}^{\varepsilon,\beta/4}$, nodes can simulate the protocol $\Pi$ with input functions $f_i(x) = \beta_0\|x - \tau(b_i)\|_2^2$. By the properties of quadratic functions, we have $F(x) = \sum_{i=1}^N f_i(x) = \beta\|x - x^*\|_2^2 + C$, where $x^* = \sum_{i=1}^N \frac{\tau(b_i)}{N}$. Thus, the output $y$ of $\Pi$ satisfies $\|y - x^*\|_2 \le (\varepsilon/\beta)^{1/2}$. The coordinator now outputs the closest point $t \in T$ to $y$. We therefore have

$$\|x^* - t\|_2 = \|x^* - y + y - t\|_2 \le \|x^* - y\|_2 + \|y - t\|_2 \le 2(\varepsilon/\beta)^{1/2} = (4\varepsilon/\beta)^{1/2}. \qquad \square$$

The next step is to prove a lower bound on the communication complexity of $\mathsf{MEAN}_{d,N}^{\varepsilon,\beta}$. We do this by using a *symmetrisation technique* of Phillips et al. [26], via reduction to the expected communication complexity of a *two-party* communication problem where one player has to learn the complete input of the other player. Specifically, in the two-player problem called $\mathsf{2\text{-}BITS}_d$, player 1 (Alice) receives a binary string $b \in \{0,1\}^d$, of length $d$, and the task is for player 2 (Bob) to output $b$. Let $\zeta_p$ be a distribution over binary strings $b \in \{0,1\}^d$ where each bit is set to 1 with probability $p$ and to 0 with probability $1 - p$. The following lower bound for $\mathsf{2\text{-}BITS}_d$ holds even for protocols with *public randomness*, i.e. when Alice and Bob have access to the same string of random bits:

**Lemma 6** ([26]). $\mathsf{ED}_{\zeta_p}^{1/3}(\mathsf{2\text{-}BITS}_d) = \Omega(dp\log p^{-1})$.

We now show that solving $\mathsf{MEAN}_{d,N}^{\varepsilon,\beta}$ requires roughly $N$ times the expected communication complexity of solving $\mathsf{2\text{-}BITS}_d$ for $d = D$ and $p = 1/2$.

**Lemma 7.** *For $N$, $d$, $\varepsilon$, and $\beta$ satisfying $d\beta/N^2\varepsilon = \Omega(1)$, we have*

$$\mathsf{RCC}^{1/3}(\mathsf{MEAN}_{d,N}^{\varepsilon,\beta}) = \Omega\big(N \cdot \mathsf{ED}_{\zeta_{1/2}}^{1/3}(\mathsf{2\text{-}BITS}_D)\big) = \Omega(Nd\log(\beta d/N\varepsilon)).$$

*Proof.* Let $\mu$ denote a distribution on $\prod_{i=1}^N \{0,1\}^D$, where each $D$-bit string is selected uniformly at random, and let $\zeta$ be uniformly random on $\{0,1\}^D$. We will prove that

$$\mathsf{D}_{\mu}^{1/3}(\mathsf{MEAN}_{d,N}^{\varepsilon,\beta}) = \Omega\big(N \cdot \mathsf{ED}_{\zeta}^{1/3}(\mathsf{2\text{-}BITS}_D)\big).$$

Since $\mathsf{ED}_{\zeta}^{1/3}(\mathsf{2\text{-}BITS}_D) = \Omega(D)$ by Lemma 6, the claim follows by Yao's Lemma.

Suppose now that we have a deterministic protocol $\Pi_1$ for $\mathsf{MEAN}_{d,N}^{\varepsilon,\beta}$ with worst-case communication cost $C$ and error probability $1/3$ on input distribution $\mu$. Given $\Pi_1$, we define a 2-player protocol $\Pi_2$ with public randomness for $\mathsf{2\text{-}BITS}_D$ as follows; assume that Alice is given $b \in \{0,1\}^D$ as input.

(1) Alice and Bob pick a random index $i \in [N]$ uniformly at to select a random $i$ node using the shared randomness. Without loss of generality, we can assume that they picked node $i = 1$.

(2) Alice and Bob simulate protocol $\Pi_1$, with Alice simulating node 1 and Bob simulating the coordinator and nodes $2, 3, \ldots, N$. For the inputs $b_1, b_2, \ldots, b_N$ to $\Pi_1$, Alice sets $b_1 = x$, and Bob selects the inputs $b_2, b_3, \ldots, b_N$ uniformly at random by using the public randomness. Messages $\Pi_1$ sends between the coordinator and node 1 are communicated between Alice and Bob, and all other communication is simulated by Bob internally.

(3) Once the simulation is complete, Bob knows the output $t \in T$ of $\Pi_1$ which satisfies $\|t - z^*\|_2 \leq (\varepsilon/\beta)^{1/2}$, where $z^* = \sum_{i=1}^{N} \tau(b_i)/N$.

As the final step, we show that Bob can now recover Alice's input from $t$. Let $y = \sum_{i=2}^{N} \tau(b_i)/(N-1)$ be the weighted average of points $\tau(b_2), \tau(b_3), \ldots, \tau(b_N)$. We now have that $N z^* - (N-1)y = \tau(b_1)$ by simple calculation.

Since $\|t - z^*\|_2 \leq (\varepsilon/\beta)^{1/2}$, it follows that

$$
\begin{aligned}
\|(Nt - (N-1)y) - \tau(b_1)\|_2 &= \|Nt - (N-1)y - Nz^* + Nz^* - \tau(b_1)\|_2 \\
&= \|Nt - Nz^* + Nz^* - (N-1)y - \tau(b_1)\|_2 \\
&= \|Nt - Nz^*\|_2 = N\|t - z^*\|_2 \leq N(\varepsilon/\beta)^{1/2} .
\end{aligned}
$$

Since the distance between any two points in $S$ is at least $3N(\varepsilon/\beta)^{1/2}$, we have that $\tau(b_1)$ is the only point from $S$ within distance $(\varepsilon/\beta)^{1/2}$ from $Nz - (N-1)y$. As Bob knows both $z$ and $\tau(b_2), \tau(b_3), \ldots, \tau(b_N)$ after the simulation, he can recover the point $x_1$ and thus infer Alice's input.

Now let us analyse the expected cost of $\Pi_2$ under input distribution $\zeta$. First, observe that since the simulation runs $\Pi_1$ on input distribution $\mu$, the output $y$ is correct with probability $2/3$, and thus the output of $\Pi_2$ is correct with probability $2/3$. Now let $C_{\Pi_1}$ be the worst-case communication cost of $\Pi_1$ and let $C_{\Pi_1}(b_1, \ldots, b_N)$ and $C_{\Pi_1,i}(b_1, \ldots, b_N)$ denote the total communication cost and the communication used by node $i$ in $\Pi_1$ on input $b_1, \ldots, b_N$, respectively. Finally, let $C_{\Pi_2}(b, r)$ be a random variable giving the communication cost of $\Pi_2$ on input $b$ and random bits $r$.

Now we have that

$$
\begin{aligned}
\mathbb{E}_{b_1, r}[C_{\Pi_2}(b_1, r)] &= \sum_{b_1 \in \{0,1\}^D} \frac{1}{2^D} \mathbb{E}_r[C_{\Pi_2}(b_1, r)] = \sum_{b_1 \in \{0,1\}^D} \frac{1}{2^D} \sum_{b_2, \ldots, b_N} \sum_{i=1}^{N} \frac{C_{\Pi_1,i}(b_1, \ldots, b_N)}{N 2^{(N-1)D}} \\
&= \frac{1}{N} \sum_{b_1, b_2, \ldots, b_N} \frac{1}{2^{ND}} \sum_{i=1}^{N} C_{\Pi_1,i}(b_1, \ldots, b_N) \\
&= \frac{1}{N} \sum_{b_1, b_2, \ldots, b_N} \frac{1}{2^{ND}} C_{\Pi_1}(b_1, b_2, \ldots, b_N) \leq \frac{1}{N} \sum_{b_1, b_2, \ldots, b_N} \frac{1}{2^{ND}} C_{\Pi_1} = \frac{C_{\Pi_1}}{N} .
\end{aligned}
$$

Since $\mathbb{E}_{b_1, \beta}[C_{\Pi_2}(b_1, r)] \geq \mathsf{ED}_\zeta^{1/3}(\text{2-BITS}_D)$, and the argument holds for any protocol $\Pi_1$ solving $\text{MEAN}_{d,N}^{\varepsilon,\beta}$ with error probability $1/3$, we have that $\mathsf{D}_\mu^{1/3}(\text{MEAN}_{d,N}^{\varepsilon,\beta}) \geq N \cdot \mathsf{ED}_\zeta^{1/3}(\text{2-BITS}_D)$, completing the proof. $\qquad\square$

Theorem 3 now follows immediately from Lemmas 5 and 7. The result can be generalised for arbitrary convex domains $\mathbb{D} \subseteq \mathbb{R}^d$ as $\Omega(N \log s)$, given a point packing bound $s$ for $\mathbb{D}$ as in Lemma 2.

## 5  Deterministic lower bound

While there remains a small gap between our main lower bound of Theorem 3 and the deterministic quantised gradient descent of Section 6, we can show that the gap cannot be closed by deterministic algorithms where the coordinator learns the value of objective function in addition to the minimiser

$x$. That is, our quantised gradient descent is the communication-optimal deterministic algorithm for problem (1) for objectives with constant condition number.

The exact result, whose proof can be found in the Appendix A, is as follows.

**Theorem 8.** *Given parameters $N$, $d$, $\varepsilon$, $\beta_0$ and $\beta = \beta_0 N$ satisfying $d\beta/\varepsilon = \Omega(1)$, any deterministic protocol solving (1) for quadratic input functions $x \mapsto \beta_0 \|x - x_0\|_2^2$ has communication complexity $\Omega(Nd\log(\beta d/\varepsilon))$, if the coordinator is also required to output estimate $r \in \mathbb{R}$ for the minimum function value such that $\sum_{i=1}^{N} f_i(z) \leq r \leq \sum_{i=1}^{N} f_i(z) + \varepsilon$.*

## 6   Communication-optimal quantised gradient descent

We now describe in detail our deterministic upper bound. Our algorithm uses quantised gradient descent, loosely following the outline of Magnússon et al. [25]. However, there are two crucial differences. First, we use a carefully-calibrated instance of the quantisation scheme of Davies et al. [11] to remove a $\log d$ factor from the communication cost, and second, we use use two-step quantisation to avoid all-to-all communication.

**Preliminaries on gradient descent.** We will assume that the input functions $f_i \colon [0,1]^d \to \mathbb{R}$ are $\alpha_0$-strongly convex and $\beta_0$-strongly smooth. This implies that $F = \sum_{i=1}^{N} f_i$ is $\alpha$-strongly convex and $\beta$-strongly smooth for $\alpha = N\alpha_0$ and $\beta = N\beta_0$. Consequently, the functions $f_i$ and $F$ have condition number bounded by $\kappa = \beta/\alpha$.

*Gradient descent* optimises the sum $\sum_{i=1}^{N} f_i(x)$ by starting from an arbitrary point $x^{(0)} \in [0,1]^d$, and applying the update rule

$$x^{(t+1)} = x^{(t)} - \gamma \sum_{i=1}^{N} \nabla f_i(x^{(t)}),$$

where $\gamma > 0$ is a parameter. It is well-known, e.g. [8], that GD converges at an exponential rate in $(\kappa - 1)/(\kappa + 1)$ for step size $\gamma = 2/(\alpha + \beta)$.

**Preliminaries on quantisation.** For compressing the gradients the nodes will send to coordinator, we use the recent quantisation scheme of Davies et al. [11]. Whereas the original uses randomised selection of the quantisation point to obtain a unbiased estimator, we can use a deterministic version that picks an arbitrary feasible quantisation point (e.g. the closest one). This gives the following guarantees:

**Corollary 9** ([11])**.** *Let $R$ and $\varepsilon$ be fixed positive parameters, and $q \in \mathbb{R}^d$ be an estimate vector, and $B \in \mathbb{N}$ be the number of bits used by the quantisation scheme. Then, there exists a deterministic quantisation scheme, specified by a function $Q_{\varepsilon,R} \colon \mathbb{R}^d \times \mathbb{R}^d \to \mathbb{R}^d$, an encoding function $\mathrm{enc}_{\varepsilon,R} \colon \mathbb{R}^d \to \{0,1\}^B$, and a decoding function $\mathrm{dec}_{\varepsilon,R} \colon \mathbb{R}^d \times \{0,1\}^B \to \mathbb{R}^d$, with the following properties:*

*(1) (Validity.) $\mathrm{dec}_{\varepsilon,R}(q, \mathrm{enc}_{\varepsilon,R}(x)) = Q_{\varepsilon,R}(x,q)$ for all $x, q \in \mathbb{R}^d$ with $\|x - q\|_2 \leq R$.*

*(2) (Accuracy.) $\|Q_{\varepsilon,R}(x,q) - x\|_2 \leq \varepsilon$ for all $x, q \in \mathbb{R}^d$ with $\|x - q\|_2 \leq R$.*

*(3) (Cost.) If $\varepsilon = \lambda R$ for any $\lambda < 1$, the bit cost of the scheme satisfies $B = O(d \log \lambda^{-1})$.*

**Algorithm description.** We now describe the algorithm, and overview its guarantees. The full description and analysis are available in Appendix C.

We assume that the constants $\alpha$ and $\beta$ are known to all nodes, so the parameters of the quantised gradient descent can be computed locally, and use $W$ to be an upper bound on the diameter on the convex domain $\mathbb{D}$, e.g. $W = d^{1/2}$ if $\mathbb{D} = [0,1]^d$. We assume that the initial iterate $x^{(0)}$ is arbitrary, but the same at all nodes, and set the initial quantisation estimate $q_i^{(0)}$ at each $i$ as the origin.

We define the following parameters for the algorithm. Let $\gamma = 2/(\alpha + \beta)$ and $\xi = \frac{\kappa-1}{\kappa+1}$ be the step size and convergence rate of gradient descent, and let $W$ be such that $\|x^{(0)} - x^*\| \leq W$. We define

$$\mu = 1 - \frac{1}{\kappa + 1}, \qquad \delta = \xi(1 - \xi)/4, \qquad R^{(t)} = \frac{2\beta}{\xi}\mu^t W,$$

where $\mu$ will be the convergence rate of our quantised gradient descent, and $\delta$ and $R^{(t)}$ will be parameters controlling the quantisation at each step. For the purposes of analysis, we assume that $\kappa \geq 2$. Note that this implies that $1/3 \leq \xi < 1$, $\mu < 1$, and $0 < \delta < 1$.

The algorithm proceeds in rounds $t = 1, 2, \ldots, T$. At the beginning of round $t + 1$, each node $i$ knows the values of the iterate $x^{(t)}$, the global quantisation estimate $q^{(t)}$, and its local quantisation estimate $q_i^{(t)}$ for $i = 1, 2, \ldots, N$. At step $t$, nodes perform the following steps:

(1) Each node $i$ updates its iterate as $x^{(t+1)} = x^{(t)} - \gamma q^{(t)}$.

(2) Each node $i$ computes its local gradient over $x^{(t+1)}$, and transmits it in quantised form to the coordinator as follows. Let $\varepsilon_1 = \delta R^{(t+1)}/2N$ and $\rho_1 = R^{(t+1)}/N$.

    (a) Node $i$ computes $\nabla f_i(x^{(t+1)})$ locally, and sends message $m_i = \mathrm{enc}_{\varepsilon_1, \rho_1}(\nabla f_i(x^{(t+1)}))$ to the coordinator.

    (b) The coordinator receives messages $m_i$ for $i = 1, 2, \ldots, N$, and decodes them as $q_i^{(t+1)} = \mathrm{dec}_{\varepsilon_1, \rho_1}(q_i^{(t)}, m_i)$. The coordinator then computes $r^{(t+1)} = \sum_{i=1}^{N} q_i^{(t+1)}$.

(3) The coordinator sends the quantised sum of gradients to all other nodes as follows. Let $\varepsilon_2 = \delta R^{(t+1)}/2$ and $\rho_2 = 2R^{(t+1)}$.

    (a) The coordinator sends the message $m = \mathrm{enc}_{\varepsilon_2, \rho_2}(r^{(t+1)})$ to each node $i$.

    (b) Each node decodes the coordinator's message as $q^{(t+1)} = \mathrm{dec}_{\varepsilon_2, \rho_2}(q^{(t)}, m)$.

After round $T$, all nodes know the final iterate $x^{(T)}$.

**Guarantees.** The key technical trick behind the algorithm is the extremely careful choice of parameters for quantisation at every step. This balances the fact that the quantisation has to be fine enough to ensure optimal GD convergence, but coarse enough to ensure optimal communication cost. Overall, the algorithm ensures the following guarantees, whose proof is provided in Appendix C.

**Theorem 10.** *Let $\varepsilon > 0$, a dimension $d$, and a convex domain $\mathbb{D} \subseteq \mathbb{R}^d$ of diameter $W$ be fixed. Given $N$ nodes, each assigned a function $f_i \colon \mathbb{D} \to \mathbb{R}$ such that $F = \sum_{i=1}^{N} f_i$ is $\alpha$-strongly convex and $\beta$-smooth, the above algorithm converges to a point $x^{(T)}$ with $F(x^{(T)}) \leq F(x^*) + \varepsilon$ using*

$$O\Big(Nd\kappa \log \kappa \log \frac{\beta W}{\varepsilon}\Big) \text{ bits of communication.}$$

## 7  Discussion and future work

We have provided the first tight bounds on the communication complexity of optimising sums of quadratic functions in the $N$-party model with a coordinator. Our results are algorithm-independent, and immediately imply the same lower bound for the practical parameter server and decentralised models of distributed optimisation.

In terms of future work, we expect that the randomised lower bound could be improved to match the deterministic one even for small $d$, possibly via reduction from a suitable *gap problem* in communication complexity (e.g. Chakrabarti and Regev [9]). Another avenue for future work is to investigate tight upper and lower bounds in the case where the functions being optimised are not quadratics, as isolating the "right" dependency on the condition number does not appear immediate. The recent results of [1, 15] suggest that the dependency on the condition number may be quite small, and therefore hard to capture without explicit limitations on the algorithm. Finally, understanding the exact complexity of optimisation in the *broadcast model*, where each message sent is seen by all nodes, and the complexity is measured by the number of bits sent, remains open.

## Acknowledgments and Disclosure of Funding

We thank the NeurIPS reviewers for insightful comments that helped us improve the positioning of our results, as well as for pointing out the subsampling approach for complementing the randomised lower bound. We also thank Foivos Alimisis and Peter Davies for useful discussions.

This project has received funding from the European Research Council (ERC) under the European Union's Horizon 2020 research and innovation programme (grant agreement No 805223 ScaleML).

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
