# A    Deterministic lower bound

While there remains a small gap between our main lower bound of Theorem 3 and the deterministic quantised gradient descent of Section 6, we can show that the gap cannot be closed by improved deterministic algorithms where the coordinator learns value of objective function $F(x)$ in addition to the minimiser $x$. That is, our quantised gradient descent is the communication-optimal deterministic algorithm for variant (1) for objectives with constant condition number.

Recall that in the $N$-player equality over universe of size $d$, denoted by $\mathsf{EQ}_{d,N}$, each player $i$ is given an input $b_i \in \{0,1\}^d$, and the task is to decide if all players have the same input. That is, $\mathsf{EQ}_{d,N}(b_1, \ldots, b_N) = 1$ if all inputs are equal, and 0 otherwise. It is known [33] that the deterministic communication complexity of $\mathsf{EQ}_{d,N}$ is $\mathsf{CC}(\mathsf{EQ}_{d,N}) = \Omega(Nd)$.

**Theorem 8.** *Given parameters $N$, $d$, $\varepsilon$, $\beta_0$ and $\beta = \beta_0 N$ satisfying $d\beta/\varepsilon = \Omega(1)$, any deterministic protocol solving (1) for quadratic input functions $x \mapsto \beta_0 \|x - x_0\|_2^2$ has communication complexity $\Omega(Nd\log(\beta d/\varepsilon))$, if the coordinator is also required to output estimate $r \in \mathbb{R}$ for the minimum function value such that $\sum_{i=1}^{N} f_i(z) \leq r \leq \sum_{i=1}^{N} f_i(z) + \varepsilon$.*

*Proof.* Assume $\Pi$ is a deterministic protocol solving (1) with communication complexity $C_\Pi$. We show that $\Pi$ can then solve $N$-party equality over a universe of size $D = \Omega(d\log(\beta d/\varepsilon))$, implying

$$C_\Pi = \Omega(ND) = \Omega(Nd\log(\beta d/\varepsilon)).$$

More specifically, let $S$ be the set given by Lemma 2 with $\delta = (2\varepsilon/\beta)^{1/2}$, and let $D = \lceil \log|S| \rceil = \Theta(d\log(\beta d/\varepsilon))$. Note that since we assume $d\beta/\varepsilon = \Omega(1)$, the set $S$ has at least two elements and $D \geq 1$. For technical convenience, assume $|S| = 2^D$, and identify each binary string $b \in \{0,1\}^D$ with an element $\tau(b) \in S$.

Next, assume that each node $i$ is given a binary string $b_i \in \{0,1\}^D$ as input, and we want to compute $\mathsf{EQ}_{D,N}(b_1, b_2, \ldots, b_N)$. The nodes simulate protocol $\Pi$ with input function $f_i$ for node $i$, where $f_i(x) = \beta_0 \|x - \tau(b_i)\|_2^2$. Let us denote $F = \sum_{i=1}^{d} f_i$. Upon termination of the protocol, the coordinator learns a point $y \in [0,1]^d$ satisfying $F(y) \leq F(x^*) + \varepsilon$ and an estimate $r \in \mathbb{R}$ satisfying $r \leq F(y) + \varepsilon$, where $x^*$ is the true global minimum. The coordinator can now adjudicate equality based on $F(y)$ as follows:

(1) If all inputs $b_i$ are equal, then the functions $f_i$ are also equal, and $F(x^*) = 0$. In this case, we have $F(y) \leq 2\varepsilon$, and the coordinator outputs 1.

(2) If there are nodes $i$ and $j$ such that $i \neq j$, then for all points $x \in [0,1]^d$, we have $f_i(x) + f_j(x) > 2\varepsilon$ by the definition of $S$, and thus $F(x^*) > 2\varepsilon$. In this case, we have $r > 2\varepsilon$, and the coordinator outputs 0.

Since communication is only used for the simulation of $\Pi$, this computes $\mathsf{EQ}_{D,N}(b_1, b_2, \ldots, b_N)$ with $C_\Pi$ total communication, completing the proof. $\qquad\square$

# B    Lower bound for non-convex functions

We now show a simple lower bound for optimisation over non-convex objective functions. We reduce from the $N$-player *set disjointness* over universe of size $d$, denoted by $\mathsf{DISJ}_{d,N}$: each player $i$ is given an input $b_i \in \{0,1\}^d$, and the coordinator needs to output 0 if there is a coordinate $\ell \in [d]$ such that $b_i(\ell) = 1$ for all $i \in [N]$, and 1 otherwise.

**Theorem 11** ([6]). *For $\delta > 0$, $N \geq 1$ and $d = \omega(\log N)$, the randomised communication complexity of set disjointness is $\mathsf{RCC}^\delta(\mathsf{DISJ}_{d,N}) = \Omega(Nd)$.*

Again consider for fixed $\varepsilon$, $d$ and $\beta$ the set $S$ given by Lemma 2 with $\delta = 2\varepsilon/\beta$. This gives a set $S$ with size at least $(\beta d^{1/2}/2C\varepsilon)^d = \exp(\Omega(d\log(\beta d)/\varepsilon)$. Let us identify the points in $S$ with indices in $[|S|]$. For a binary string $b \in \{0,1\}^{|S|}$, define the function $f_b$ by

$$f_b(x) = \begin{cases} \beta\|x - s\|_2 & \text{if } \|x - s\|_2 < \varepsilon/\beta \text{ for } s \text{ with } b_s = 1, \\ \varepsilon & \text{otherwise.} \end{cases}$$

Since the distance between points in $S$ is at least $2\varepsilon/\beta$, the functions $f_T$ are well-defined, continuous and $\beta$-Lipschitz.

**Theorem 12.** *Given parameters $N$, $d$, $\varepsilon$ and $\beta$ satisfying $d\beta/\varepsilon = \Omega(1)$ and $(\beta d^{1/2}/2C\varepsilon)^d = \omega(\log N)$, any protocol solving 1 with error probability $\delta > 0$ when the inputs are guaranteed to be functions $f_b$ for $b \in \{0,1\}^{|S|}$ has communication complexity $N\exp(\Omega(d\log(\beta d)/\varepsilon))$.*

*Proof.* Assume there is a protocol $\Pi$ with the properties stated in the claim, and worst-case communication cost $C_\Pi$. We now show that we can use $\Pi$ to solve set disjointness over universe of size $|S|$ with $C_\Pi$ total communication, which implies

$$C_\Pi \geq \mathsf{RCC}^\delta(\mathsf{DISJ}_{|S|,N}) = \Omega(N\exp(\Omega(d\log(\beta d)/\varepsilon))),$$

yielding the claim.

First, we note that after running $\Pi$, the coordinator can send the final estimate $z$ of the optimum to all nodes, and receive approximations of the local function values $f_i(z)$ from all nodes with additive $O(Nd\log d\beta/\varepsilon)$ overhead, e.g. using quantisation of Corollary 9. We can without loss of generality assume that this does not exceed the total communication cost of $\Pi$.

For $b_1, b_2, \ldots b_N \in \{0,1\}^{|S|}$ that all contain 1 in some position $s$, then we have $\sum_{i=1}^{N} f_{b_i}(x) = 0$. Otherwise, for any point $x \in [0,1]^d$, consider the closest point $s \in S$ to $x$; there is at least one $b_i$ with $b_s = 0$, and for that function $f_{b_i}(x) = \varepsilon$ by definition. Thus, if $b_1, b_2, \ldots b_N$ are a YES-instance for set disjointness, then $\inf_{x \in [0,1]^d} \sum_{i=1}^{N} f_{b_i}(x) \geq \varepsilon$, and if $b_1, b_2, \ldots b_N$ are a NO-instance, then $\inf_{x \in [0,1]^d} \sum_{i=1}^{N} f_{b_i}(x) = 0$.

By definition, $\Pi$ can be used to distinguish between the two cases, and thus to solve set disjointness. $\square$

# C   Communication-optimal quantised gradient descent, full version

We now describe in detail our deterministic upper bound. Our algorithm uses quantised gradient descent, loosely following the outline of Magnússon et al. [25]. However, there are two crucial differences. First, we use a carefully-calibrated instance of the quantisation scheme of Davies et al. [11] to remove a $\log d$ factor from the communication cost, and second, we use use two-step quantisation to avoid all-to-all communication.

**Preliminaries on gradient descent.** We will assume that the input functions $f_i\colon [0,1]^d \to \mathbb{R}$ are $\alpha_0$-strongly convex and $\beta_0$-strongly smooth. This implies that $F = \sum_{i=1}^{N} f_i$ is $\alpha$-strongly convex and $\beta$-strongly smooth for $\alpha = N\alpha_0$ and $\beta = N\beta_0$. Consequently, the functions $f_i$ and $F$ have condition number bounded by $\kappa = \beta/\alpha$.

*Gradient descent* optimises the sum $\sum_{i=1}^{N} f_i(x)$ by starting from an arbitrary point $x^{(0)} \in [0,1]^d$, and applying the update rule

$$x^{(t+1)} = x^{(t)} - \gamma \sum_{i=1}^{N} \nabla f_i(x^{(t)}),$$

where $\gamma > 0$ is a parameter.

Let $x^*$ denote the global minimum of $F$. We use the following standard result on the convergence of gradient descent; see e.g. Bubeck [8].

**Theorem 13.** *For $\gamma = 2/(\alpha + \beta)$, we have that $\|x^{(t+1)} - x^*\|_2 \leq \frac{\kappa-1}{\kappa+1}\|x^{(t)} - x^*\|_2$.*

**Preliminaries on quantisation.** For compressing the gradients the nodes will send to coordinator, we use the recent quantisation scheme of Davies et al. [11]. Whereas the original uses randomised selection of the quantisation point to obtain a unbiased estimator, we can use a deterministic version that picks an arbitrary feasible quantisation point (e.g. the closest one). This gives the following guarantees:

**Corollary 14** ([11])**.** *Let $R$ and $\varepsilon$ be fixed positive parameters, and $q \in \mathbb{R}^d$ be an estimate vector, and $B \in \mathbb{N}$ be the number of bits used by the quantisation scheme. Then, there exists a deterministic quantisation scheme, specified by a function $Q_{\varepsilon,R}\colon \mathbb{R}^d \times \mathbb{R}^d \to \mathbb{R}^d$, an encoding function*

$\text{enc}_{\varepsilon,R} \colon \mathbb{R}^d \to \{0,1\}^B$, *and a decoding function* $\text{dec}_{\varepsilon,R} \colon \mathbb{R}^d \times \{0,1\}^B \to \mathbb{R}^d$, *with the following properties:*

*(1) (Validity.)* $\text{dec}_{\varepsilon,R}(q, \text{enc}_{\varepsilon,R}(x)) = Q_{\varepsilon,R}(x, q)$ *for all* $x, q \in \mathbb{R}^d$ *with* $\|x - q\|_2 \leq R$.

*(2) (Accuracy.)* $\|Q_{\varepsilon,R}(x, q) - x\|_2 \leq \varepsilon$ *for all* $x, q \in \mathbb{R}^d$ *with* $\|x - q\|_2 \leq R$.

*(3) (Cost.) If* $\varepsilon = \lambda R$ *for any* $\lambda < 1$*, the bit cost of the scheme satisfies* $B = O(d \log \lambda^{-1})$.

## C.1 Algorithm description

We now describe the algorithm, and overview its guarantees. We assume that the constants $\alpha$ and $\beta$ are known to all nodes, so the parameters of the quantised gradient descent can be computed locally, and use $W$ to be an upper bound on the diameter on the convex domain $\mathbb{D}$, e.g. $W = d^{1/2}$ if $\mathbb{D} = [0,1]^d$. We assume that the initial iterate $x^{(0)}$ is arbitrary, but the same at all nodes, and set the initial quantisation estimate $q_i^{(0)}$ at each $i$ as the origin.

We define the following parameters for the algorithm. Let $\gamma = 2/(\alpha + \beta)$ and $\xi = \frac{\kappa - 1}{\kappa + 1}$ be the step size and convergence rate of gradient descent, and let $W$ be such that $\|x^{(0)} - x^*\| \leq W$. We define

$$\mu = 1 - \frac{1}{\kappa + 1}, \qquad \delta = \xi(1 - \xi)/4, \qquad R^{(t)} = \frac{2\beta}{\xi} \mu^t W,$$

where $\mu$ will be the convergence rate of our quantised gradient descent, and $\delta$ and $R^{(t)}$ will be parameters controlling the quantisation at each step. For the purposes of analysis, we assume that $\kappa \geq 2$. Note that this implies that $1/3 \leq \xi < 1$, $\mu < 1$, and $0 < \delta < 1$.

The algorithm proceeds in rounds $t = 1, 2, \ldots, T$. At the beginning of round $t + 1$, each node $i$ knows the values of the iterate $x^{(t)}$, the global quantisation estimate $q^{(t)}$, and its local quantisation estimate $q_i^{(t)}$ for $i = 1, 2, \ldots, N$. At step $t$, nodes perform the following steps:

(1) Each node $i$ updates its iterate as $x^{(t+1)} = x^{(t)} - \gamma q^{(t)}$.
(2) Each node $i$ computes its local gradient over $x^{(t+1)}$, and transmits it in quantised form to the coordinator as follows. Let $\varepsilon_1 = \delta R^{(t+1)}/2N$ and $\rho_1 = R^{(t+1)}/N$.
    (a) Node $i$ computes $\nabla f_i(x^{(t+1)})$ locally, and sends message $m_i = \text{enc}_{\varepsilon_1,\rho_1}(\nabla f_i(x^{(t+1)}))$ to the coordinator.
    (b) The coordinator receives messages $m_i$ for $i = 1, 2, \ldots, N$, and decodes them as $q_i^{(t+1)} = \text{dec}_{\varepsilon_1,\rho_1}(q_i^{(t)}, m_i)$. The coordinator then computes $r^{(t+1)} = \sum_{i=1}^{N} q_i^{(t+1)}$.
(3) The coordinator sends the quantised sum of gradients to all other nodes as follows. Let $\varepsilon_2 = \delta R^{(t+1)}/2$ and $\rho_2 = 2R^{(t+1)}$.
    (a) The coordinator sends the message $m = \text{enc}_{\varepsilon_2,\rho_2}(r^{(t+1)})$ to each node $i$.
    (b) Each node decodes the coordinator's message as $q^{(t+1)} = \text{dec}_{\varepsilon_2,\rho_2}(q^{(t)}, m)$.

After round $T$, all nodes know the final iterate $x^{(T)}$.

## C.2 Analysis

For simplicity, we will split the analysis into two parts. The first describes and analyses the algorithm in an abstract way; the second part describes the details of implementing it in the coordinator model. For technical convenience, assume $\kappa \geq 2$; for smaller condition numbers, we can run the algorithm with $\kappa = 2$.

**Convergence.** Let $\gamma = 2/(\alpha + \beta)$, let $x^{(0)} \in [0,1]^d$, $q^{(0)} \in \mathbb{R}^d$ and $q_i^{(0)} \in \mathbb{R}^d$ for $i = 1, 2, \ldots, N$ be arbitrary initial values. From the algorithm description, we see that the update rule for our quantised gradient descent is

$$x^{(t+1)} = x^{(t)} - \gamma q^{(t)},$$
$$q_i^{(t+1)} = Q_{\varepsilon_1,\rho_1}(\nabla f_i(x^{(t+1)}), q_i^{(t)}) \qquad \text{for } \varepsilon_1 = \delta R^{(t+1)}/2N \text{ and } \rho_1 = R^{(t+1)}/N,$$

$$r^{(t+1)} = \sum_{i=1}^{N} q_i^{(t+1)},$$

$$q^{(t+1)} = Q_{\varepsilon_2,\rho_2}\left(r^{(t+1)}, q^{(t)}\right) \qquad \text{for } \varepsilon_2 = \delta R^{(t+1)}/2 \text{ and } \rho_2 = 2R^{(t+1)}.$$

**Lemma 15.** *The inequalities*

$$\|x^{(t)} - x^*\|_2 \le \mu^t W, \tag{Q1}$$

$$\|\nabla f_i(x^{(t)}) - q_i^{(t)}\|_2 \le \delta R^{(t)}/2N, \tag{Q2}$$

$$\|\nabla F(x^{(t)}) - q^{(t)}\|_2 \le \delta R^{(t)} \tag{Q3}$$

*hold for all $t$, assuming that they hold for $x^{(0)}$, $q^{(0)}$ and $q_i^{(0)}$ for $i = 1, 2, \ldots, N$.*

*Proof.* We apply induction over $t$; we assume that the inequalities (Q1-Q3) hold for $t$, and prove that they also hold for $t + 1$. Since we assume the inequalities hold for $t = 0$, the base case is trivial. By elementary computation, the following hold:

$$0 < \xi < 1, \qquad 0 < \delta < 1, \qquad 2\delta/\xi + \xi = \mu, \qquad \gamma\beta \le 2, \qquad \mu R^{(t)} = R^{(t+1)}.$$

*Convergence (Q1)*: First, we observe that $\frac{2\delta}{\xi} + \xi = \frac{1}{2}(1 + \xi) = 1 - \frac{1}{\kappa+1} = \mu$ and $\gamma\beta \le 2$. We now have that

$$
\begin{aligned}
\|x^{(t+1)} - x^*\|_2 &= \|x^{(t)} - \gamma q^{(t)} + \gamma\nabla F(x^{(t)}) - \gamma\nabla F(x^{(t)}) + x^*\|_2 && \text{(def.)} \\
&\le \|\gamma q^{(t)} - \gamma\nabla F(x^{(t)})\|_2 + \|(x^{(t)} - \gamma\nabla F(x^{(t)})) - x^*\|_2 && \text{(triangle-i.e.)} \\
&\le \gamma\|\nabla F(x^{(t)}) - q^{(t)}\|_2 + \xi\|x^{(t)} - x^*\|_2 && \text{(norm, Thm. 13)} \\
&\le \gamma\delta R^{(t)} + \xi\mu^t W && \text{(by Q1, Q3 for } t) \\
&= (\gamma\beta\delta/\xi + \xi)\mu^t W && \text{(expand } R^{(t)}) \\
&\le (2\delta/\xi + \xi)\mu^t W = \mu^{t+1} W. && (\gamma\beta \le 2)
\end{aligned}
$$

*Local quantisation (Q2)*: First, let us observe that to prove that (Q2) holds for $t + 1$, it is sufficient to show $\|\nabla f_i(x^{(t+1)}) - q_i^{(t)}\|_2 \le R^{(t+1)}/N$, as the claim then follows from the definition of $q_i^{(t+1)}$ and Corollary 9. We have

$$
\begin{aligned}
\|\nabla f_i(x^{(t+1)}) - q_i^{(t)}\|_2 &= \|\nabla f_i(x^{(t+1)}) - \nabla f_i(x^{(t)}) + \nabla f_i(x^{(t)}) - q_i^{(t)}\|_2 \\
&\le \|\nabla f_i(x^{(t+1)}) - \nabla f_i(x^{(t)})\|_2 + \|\nabla f_i(x^{(t)}) - q_i^{(t)}\|_2 && \text{(triangle-i.e.)} \\
&\le \beta_0\|x^{(t+1)} - x^{(t)}\|_2 + \delta R^{(t)}/N && \text{(smoothness, Q3)} \\
&\le \beta_0\left(\|x^{(t+1)} - x^*\|_2 + \|x^{(t)} - x^*\|_2\right) + \delta R^{(t)}/N && \text{(triangle-i.e.)} \\
&\le 2\beta_0\mu^t W + \delta R^{(t)}/N && \text{(by Q1 for } t, t+1) \\
&= 2\beta\mu^t W/N + \delta R^{(t)}/N && (\beta = \beta_0 N) \\
&= \xi R^{(t)}/N + \delta R^{(t)}/N && \text{(definition of } R^{(t)}) \\
&= (\xi + \delta)R^{(t)}/N && \text{(rearrange)} \\
&\le (\xi + 2\delta/\xi)R^{(t)}/N && (2/\xi \ge 1) \\
&= \mu R^{(t)}/N = R^{(t+1)}/N. && (2\delta/\xi + \xi = \mu)
\end{aligned}
$$

*Global quantisation (Q3)*: To prove (Q3), we start by giving two auxiliary inequalities. First, we prove that $\|\nabla F(x^{(t+1)}) - r^{(t+1)}\|_2 \le \delta R^{(t+1)}/2$:

$$
\begin{aligned}
\|\nabla F(x^{(t+1)}) - r^{(t+1)}\|_2 &= \|\sum_{i=1}^{N} \nabla f_i(x^{(t+1)}) - \sum_{i=1}^{N} q_i^{(t+1)}\|_2 && \text{(def.)} \\
&\le \sum_{i=1}^{N} \|\nabla f_i(x^{(t+1)}) - q_i^{(t+1)}\|_2 && \text{(triangle-i.e.)}
\end{aligned}
$$

$$\leq N\delta R^{(t+1)}/2N = \delta R^{(t+1)}/2\,. \qquad\qquad \text{(by Q2 for } t+1)$$

Next, we want to prove $\|r^{(t+1)} - q^{(t+1)}\|_2 \leq \delta R^{(t+1)}/2$. Again, it is sufficient to show $\|r^{(t+1)} - q^{(t)}\|_2 \leq 2R^{(t+1)}$, as the claim then follows from the definition of $q^{(t+1)}$ and Corollary 9. We have

$$
\begin{aligned}
\|r^{(t+1)} - q^{(t)}\|_2 &= \|r^{(t+1)} + \nabla F(x^{(t+1)}) - \nabla F(x^{(t+1)}) + \nabla F(x^{(t)}) - \nabla F(x^{(t)}) - q^{(t)}\|_2 \\
&\leq \|r^{(t+1)} - \nabla F(x^{(t+1)})\|_2 + \|\nabla F(x^{(t+1)}) - \nabla F(x^{(t)})\|_2 + \|\nabla F(x^{(t)}) - q^{(t)}\|_2 \\
&\leq \delta R^{(t+1)}/2 + \beta\|x^{(t+1)} - x^{(t)}\|_2 + \delta R^{(t)}\,,
\end{aligned}
$$

where the last inequality follows from smoothness of $F$, equation (Q2) for $t+1$ and equation (Q3) for $t$. It holds that

$$
\begin{aligned}
\beta\|x^{(t+1)} - x^{(t)}\|_2 + \delta R^{(t)} &\leq \beta\big(\|x^{(t+1)} - x^*\|_2 + \|x^{(t)} - x^*\|_2\big) + \delta R^{(t)} && \text{(triangle-i.e.)} \\
&\leq 2\beta\mu^t W + \delta R^{(t)} && \text{(by Q1 for } t, t+1) \\
&= \xi R^{(t)} + \delta R^{(t)} && \text{(definition of } R^{(t)}) \\
&\leq (\xi + 2\delta/\xi)R^{(t)} && (2/\xi \geq 1) \\
&= \mu R^{(t)} = R^{(t+1)}\,.
\end{aligned}
$$

Combining the two previous inequalities, we have

$$\|r^{(t+1)} - q^{(t)}\|_2 \leq \delta R^{(t+1)}/2 + R^{(t+1)} \leq 2R^{(t+1)}\,,$$

as desired.

Finally, putting things together, we have

$$
\begin{aligned}
\|\nabla F(x^{(t+1)}) - q^{(t+1)}\|_2 &= \|\nabla F(x^{(t+1)}) - r^{(t+1)} + r^{(t+1)} - q^{(t+1)}\|_2 \\
&\leq \|\nabla F(x^{(t+1)}) - r^{(t+1)}\|_2 + \|r^{(t+1)} - q^{(t+1)}\|_2 \\
&\leq \delta R^{(t+1)}/2 + \delta R^{(t+1)}/2 = \delta R^{(t+1)}\,,
\end{aligned}
$$

completing the proof. $\qquad\square$

**Lemma 16.** *For any $\varepsilon > 0$ and $t \geq (\kappa+1)\log\frac{W}{\varepsilon}$, we have $\|x^{(t)} - x^*\|_2 \leq \varepsilon$.*

*Proof.* By Lemma 15, we have $\|x^{(t)} - x^*\|_2 \leq \mu^t W = (1 - (1-\mu))^t W \leq e^{-(1-\mu)t}W$. Assuming $t \geq \frac{1}{1-\mu}\log\frac{W}{\varepsilon}$, we have

$$e^{-(1-\mu)t}W \leq e^{-(1-\mu)(1-\mu)^{-1}\log W/\varepsilon}W = e^{\log\varepsilon/W}W = \varepsilon W/W = \varepsilon\,.$$

The claim follows by observing that $\frac{1}{1-\mu} = \kappa + 1$ by definition. $\qquad\square$

**Communication cost.** Finally, we analyse the distributed implementation described at the beginning of this section, and analyse its total communication cost. Recall that we assume that the parameters $\alpha$ and $\beta$ are known to all nodes, so the parameters of the quantised gradient descent can be computed locally, and use $W = d^{1/2}$. Note that $W$ is the only parameter depending on the input domain, so the algorithm also applies for arbitrary convex domain $\mathbb{D} \subseteq \mathbb{R}^d$, setting $W$ to be the diameter of $\mathbb{D}$.

Since $\delta < 1$, we have by Lemma 9 that the each of the messages sent by the nodes has length at most $O(d\log\delta^{-1})$ bits. Assuming $\kappa \geq 2$, we have $\xi \geq 1/3$ and

$$\log\delta^{-1} = \log\frac{2(\kappa+1)}{\xi} \leq \log 6(\kappa+1) \leq \log 7\kappa\,.$$

Since the nodes send a total of $2N$ messages of $O(d\log\kappa)$ bits each, the total communication cost of a single round is $O(Nd\log\kappa)$ bits.

To get $F(x^{(T)}) - F(x^*) \leq \varepsilon$, we need $\|x^{(T)} - x^*\|_2 \leq (\varepsilon/\beta)^2$. By Lemma 16, selecting $T = O(\kappa\log\frac{\beta W}{\varepsilon})$ is sufficient. Finally, using $W = O(d^{1/2})$, we have that the total communication cost of the optimisation is $O\big(Nd\kappa\log\kappa\log\frac{\beta d}{\varepsilon}\big)$.

## D  Subsampling

In this section, we show that the condition $\beta d/N^2\varepsilon = \Omega(1)$ in our main lower bound is, to a degree, necessary.

**Lemma 17.** *Let $S = \{x_1, x_2, \ldots, x_N\} \subseteq [0,1]^d$, and let $X_1, X_2, \ldots, X_M$ be i.i.d. random variables, each selected uniformly at random from $S$. Writing $\hat{x} = \frac{1}{N}\sum_{i=1}^{N} x_i$ and $X = \frac{1}{M}\sum_{i=1}^{M} X_i$, we have*

$$\mathbb{E}\big[\|X - \hat{x}\|_2\big] = 0, \qquad \text{and} \qquad \mathrm{Var}\big(\|X - \hat{x}\|_2\big) = \frac{d}{M}.$$

*Proof.* The first part follows immediately by the definition of the expectation. For the second part, we first note that since all points within $[0,1]^d$ are at most $d^{1/2}$ apart, and thus by the definition of variance, it follows that

$$\mathrm{Var}(\|X - \hat{x}\|_2) = \mathbb{E}[\|X - \hat{x}\|_2^2] - \mathbb{E}[\|X - \hat{x}\|_2]^2 = \mathbb{E}[\|X - \hat{x}\|_2^2] - 0$$

$$\leq \mathbb{E}\Big[\frac{1}{M^2}\sum_{i=1}^{M}\|X_i - \hat{x}\|_2^2\Big] = \frac{1}{M^2}\sum_{i=1}^{M}\mathbb{E}\big[\|X_i - \hat{x}\|_2^2\big]$$

$$\leq \frac{1}{M^2}Md = \frac{d}{M}.$$

$\square$

**Theorem 18.** *Assume the input functions $f_i$ of the nodes are promised to be quadratic functions $x \mapsto \beta_0\|x - x^*\|_2^2$ for some constant $\beta_0 > 0$, let $\beta = \beta_0 N$, and assume we can select $M \leq N$ to be an integer satisfying $\beta d/M\varepsilon \leq 1/8$. The there is a randomised algorithm solving (1) using*

$$O\Big(Md \log \frac{\beta d}{\varepsilon}\Big) \text{ bits of communication,}$$

*with probability at least $1/2$.*

*Proof.* We start by having the coordinator select a multiset $I$ of $M$ nodes uniformly at random with replacement. Let $\hat{x}$ denote the global optimum of $\sum_{i=1}^{N} f_i$, and let $\hat{Y}$ be the random variable for the global optimum of $\sum_{i \in I} f_i$. By Lemma 17, Chebyshev's inequality and the assumption $\beta d/N\varepsilon \leq 1/8$, we have that

$$\Pr\Big[\|\hat{Y} - \hat{x}\|_2 \geq \frac{1}{2}\Big(\frac{\varepsilon}{\beta}\Big)^{1/2}\Big] \leq \frac{4d\beta}{\varepsilon M} \leq 1/2.$$

Let $\hat{y}$ be the actualised value of $\hat{Y}$. We now apply the algorithm of Theorem 10 to find a point $z$ such that $\|z - \hat{y}\|_2 \leq 1/2(\varepsilon/\beta)^{1/2}$, where, if the multiset $I$ contains duplicates, those nodes simulate multiple copies of themselves. This uses $O(Md \log \beta d/\varepsilon)$ bits of communication. We now have with probability at least $1/2$ that $\|z - \hat{x}\| \leq (\varepsilon/\beta)^{1/2}$, and thus $\sum_{i=1}^{N} f_i(z) \leq \sum_{i=1}^{N} f_i(\hat{x}) + \varepsilon$. $\square$