# OpenReview forum: "Towards Tight Communication Lower Bounds for Distributed Optimisation"
_NeurIPS.cc/2021/Conference — NeurIPS 2021 Poster_

### Official Review · Reviewer_coZz · 2021-07-15

**Rating:** 6
**Confidence:** 3

**Summary:**

The submission studies the theoretical coordinator model communication complexity of multiple parties, each holding a strongly convex f_i, for the task of approximately minimizing the sum f_i. The main result is a communication lower bound, which basically follows by picking the approximation parameter so restrictive that each party has to send the minimum of its f_i, to appropriate precision, in all dimensions.

**Main Review:**

Communication complexity is an important bottleneck for optimization. I am not excited by the lower bound. I am less of an expert about the algorithm side – to me their approach seems natural but not trivial. This work already inspired an ICML’21 paper that develops a more interesting algorithm.


Lower bound parameters: the lower bound is based on the sum of quadratics || x – x_i ||^2_2, where each party holds one x_i. The lower bound holds for approximation \epsilon = O(d / N), where d = dimension, N = # of parties. The dependence on d is the natural one and is technically dealt with an off-the-shelf packing bound (any good binary error correcting code would do). The dependence on N is what makes the lower bound not exciting IMHO: with \epsilon ~ 1/N^2 per dimension and party, this obviously forces all the parties to communicate. This result may be tight in this regime, but it’s probably not so natural – on say a random input the value of the function would be on the order of magnitude of d*N, so in comparison this is a really small \epsilon. In practice, for the sum of quadratics considered in the lower bound, taking a sub-sample of the parties’ functions would probably be good enough for most applications where N is so large that we’re actually concerned about an asymptotic factor of N in the communication.


Typos:
Line 241: “will prove” -> “we will prove”
Line 242: \ref -> \eqref


**Time Spent Reviewing:**

3

---

> ### Author Response · Authors · 2021-08-10
> **Response to Reviewer coZz**
>
> > Lower bound parameters:
>
> For context, it is worth comparing our main lower bound with the two-node setting, where the lower bound $\Omega(d \log \beta d /\varepsilon)$ is known to be tight. The "expected" lower bound for the N-node setting would be $\Omega(N d \log \beta d /\varepsilon)$, but so far we have not been able to prove this without additional assumptions.
>
> Our main lower bound of $\Omega(N d \log \beta d / N \varepsilon)$ does indeed require additional coupling of the parameters due to the constraint $\beta d/ N^2 \varepsilon = \Omega(1)$, and this constraint is required for the bound to even make sense (otherwise the log is negative.) However, there are practically relevant regimes in which $\varepsilon$ does not need to depend directly on $N$, e.g. if we consider setting with $d = \Omega(N^{2+\delta})$ for $\delta > 0$.
> We believe that this setting is reasonable, as with modern ML tasks,  $d$ can be easily in millions, and the number of machines is at most in the thousands.
>
> > For the sum of quadratics considered in the lower bound, subsampling would suffice
>
> This is a valid point, but notice that our lower bound extends to more complex function families, and in particular can be further boosted for arbitrary non-convex functions.

---

> > ### Comment · Reviewer_coZz · 2021-08-13
> > **Subsampling**
> >
> > (1) The natural way to solve the instance in your lower bound is subsampling; and
> > (2) This subsampling is ruled out by your choice of parameters.
> > To me this suggests that the problem is with the choice of parameters.
> >
> > (Of course your lower bound extends to more complex functions - my point is that if you try to argue that this easy function is hard in a certain sense, then this notion of hardness isn't very interesting.)

---

> > > ### Author Response · Authors · 2021-08-18
> > > **Re. Subsampling**
> > >
> > > We would like to thank the reviewer for the follow-up, and apologize for the slight delay in replying (due to travel).
> > >
> > > We fully agree that for "simple" problem instances, and large N, sub-sampling would indeed work. Even further, for very simple problem instances and large epsilon, agents could even solve the problem entirely locally, without any communication at all. However, we would argue that this is not that common in the distributed setting, and that, in practical settings such as in federated learning, it is often the case that the functions being learned are complex and data is scarce, so examining the entire data is necessary, and non-trivial infrastructure is being built for this purpose.
> > >
> > > Still, in this context, it is reasonable to ask whether our lower bound says anything "interesting" for simple functions. We believe the answer is yes: The only condition for the lower bound to apply in the convex case is, as stated in the paper,
> > >
> > > βd/N^2ε > πe
> > >
> > > where π and e are the classic constants. For illustration, if we set ε = 10^-5, which is common, and N in the range 10-100, which is again common, we get that the lower bound would apply to most reasonable problem instances, as βd > 1 is a fairly easy requirement to meet. Clearly, the lower bound becomes stronger if we are allowed to work with more complex objectives.
> > >
> > > We fully with the reviewer that this point should be made very clear, and will add a detailed discussion in the next revision.

---

> > > > ### Author Response · Authors · 2021-09-01
> > > > **A more detailed response**
> > > >
> > > > In light of Reviewer WQv3's message, we realized our previous response may have been too high-level, and caused confusion. We have therefore posted a more detailed response specifically focusing on the parameter dependencies required by our results, and on their interaction with sub-sampling.
> > > >
> > > > The detailed response can be found here:
> > > > https://openreview.net/forum?id=86iCmraCBL&noteId=0jpLBrFm7YM
> > > >
> > > > We include a short summary of this discussion here:
> > > >
> > > > — Essentially, the best possible lower bound one might expect in our setting is of the form $\Omega(Nd \log d / \varepsilon)$. We match this ideal result for deterministic algorithms.
> > > >
> > > > — Getting this type of "ideal" lower bound for randomised algorithms, using e.g. subsampling, *requires* restrictions on the parameters. One natural such restriction is $\varepsilon < d / N$, since when we have $\varepsilon = \omega(d / N)$, one can get $o(N d \log d/\varepsilon)$ communication for mean estimation by subsampling. This is illustrated in the detailed response linked to above.
> > > >
> > > >  — Our lower bound gives $\Omega(Nd \log d / N\varepsilon)$ and requires $\varepsilon = O(d/N^2)$, so we lose slightly in both the bound and the parameter requirements to approach randomized algorithms. However, as we show above, some parameter requirements do appear *necessary*, and obtaining truly tight bounds both in terms of bound and parameter dependencies appears challenging. Our result is the first to approach the problem at this level of generality, and it does apply to reasonable parameter values.
> > > >
> > > >  —  Our lower bound can be also interpreted as saying that any subsampling strategy must use a large sample set for certain parameters. (Again, this is illustrated in the full response.)

---

### Official Review · Reviewer_aPic · 2021-07-16

**Rating:** 6
**Confidence:** 4

**Summary:**

This paper studies the problem of distributed convex optimization, where each of $N$ machines holds a local function $f_i$, and a coordinator aims to minimize the global objective $\sum_i f_i$ by using a limited amount of communication with local machines. The main contribution of this paper is a $\tilde{\Omega}(Nd)$ bits lower bound on the total communication cost. The lower bound is constructed by considering quadratic local functions and reducing the minimization problem to mean estimation and two-party communication problems. Finally, the authors show that the lower bound can be attained when $f_i$'s are strongly convex and smooth.

**Limitations And Societal Impact:**

The authors have listed the limitations as part of future works. This work does not present any foreseeable societal consequences due to its theoretical nature.

**Main Review:**

Though the lower bound $\tilde{\Omega}(Nd)$ may be expected and not too surprising given previous works on the two-node settings, the result can still be useful in related directions such as federated learning. In general, the paper is well-written and the proofs are clear and easy to follow.  I have the following questions and suggestions:

1. The paper considers the private randomness model, but I wonder whether the same bound holds for the shared randomness settings (given that the problem is ultimately reduced to 2-BITS and the lower bound for 2-BITS holds for the shared randomness model).


2. In many distributed/federated learning scenarios, the communication is asymmetric; that is, the uplink communication (i.e. from each machine to the coordinator) is usually more expensive than the downlink communication.  Is there a lower bound for only the uplink communication cost (with an assumption that the downlink bandwidth is infinite)?


3. The authors may want to add some comments on whether the lower bound is tight for functions without strong convexity or smoothness assumptions.


4. Finally, it would be better to include some references and comparisons with the line of research that study the communication complexity under gradient oracle settings. See [1] and its references for example.


[1] Acharya et al. "Information-constrained optimization: can adaptive processing of gradients help?"

**Time Spent Reviewing:**

4

---

> ### Author Response · Authors · 2021-08-10
> **Response to Reviewer aPic**
>
> Thank you for the very interesting questions!
>
>  1. The main lower bound result should indeed work with shared randomness essentially as written. We had simply initially started with private randomness as a reasonably realistic model, and did not consider updating the rest of the paper to use shared randomness.
>
>  2. The free downlink communication model seems like an interesting variant, and it looks like the main lower should work in this setting too. That is, the 2-BITS lower bound should hold also when only communication from Alice to Bob is counted (as Bob has no input), and then one can follow the proof of Lemma 7 almost verbatim. However, details still need to be checked carefully, which we will do for the next revision. The lower bounds presented in the appendices most likely do not work in this model, at least without significant additional work; in particular, proof of Theorem 10 fails completely, as equality is not hard with unlimited downlink communication.
>
>  3. We will revise the discussion in future versions. Note that in general, removing these assumptions makes the problem harder, e.g. see Theorem 12 in the appendix for an extreme example. Understanding how the communication complexity behaves in less restrictive but still realistic setting is a direction we are interested in exploring further in future work.
>
>  4. Thank you for the reference, we were not aware of this recent work. We did not discuss oracle settings beyond the mention in the related work due to space limitations, and, furthermore, the results from oracle setting do not directly relate to communication complexity as formalised here, as there the measure of interest is the number of oracle accesses. By contrast, the communication complexity model allows arbitrary access to the local functions for free, i.e. the lower bounds work also against algorithms that perform arbitrarily many local oracle accesses. That said, understanding the tradeoffs between various resources (oracle accesses, local computation, communication) in distributed optimisation is an interesting question. In particular, one question we find fascinating is whether the lower bound can be "boosted" by limiting the oracle access to e.g. first-order information.

---

> > ### Comment · Reviewer_aPic · 2021-08-31
> > **To the authors**
> >
> > I have read the response from the authors and the reviews made by the other reviewers. I am satisfied with the response, especially points 1 and 2 regarding the shared randomness/asymmetric communication models. I suggest the authors to include some discussions on them in the next revision (even without formal proofs), which would also be useful to the community.

---

### Official Review · Reviewer_JTxy · 2021-07-16

**Rating:** 6
**Confidence:** 2

**Summary:**

In this paper, authors propose the communication complexity lower and upper bound for the sum of quadratic functions minimization in the distributed setup.

**Limitations And Societal Impact:**

yes

**Main Review:**

In this paper, author consider the communication bottleneck problem in distributed optimization algorithms.
The main result of this paper is the lower-bound on the communication complexity for the quadratic functions minimization.

This paper is pure theoretical and consider only the quadratic functions with additive error tolerance stopping criteria.
To proof the main result, authors consider discrete model with finite input and output sets.
Then they move from the initial problem to the problem of finding of the mean of the parameters that is necessary due to the explicit form of the sum of quadratic functions. This bound is sufficiently get from the complexity of the two-player game 2-BITS_d.
As a result, I find this theory quite nice but I am hesitating if this bound is useful in practice. Since the considered set up is quadratic functions, I would expect this minimization problem be easy enough to not talk about the communication bottle-neck and complexity.
However, the future plans to extend the theory to some other classes of inputs sounds nice. I would be interested to see the result for LASSO problem, where we have an additional $\ell_1$ regularizer that enforces some structure to the solution. I am wondering if this structure can somehow appear in this bound (if a-priori it is known that the solution is 1% dense).
 Moreover, as far as I understand, the lower bound is not tight at all, so I would say that the discussion about its tightness is missing here.

According to the upper-bound, I am wondering, why exactly this method is selected for it. (at least we should talk about the projected gradient descent to have $x^k\in [0,1]^d$).


**Time Spent Reviewing:**

1 hour

---

> ### Author Response · Authors · 2021-08-10
> **Response to Reviewer JTxy**
>
> > Practical significance of the bound
>
> While our immediate goal is theoretical, we believe our result does have some practical implications.
> Notice that our bound would apply in the context of large-scale distributed optimization, such as federated learning, saying that essentially, in the worst case, the server has to scale its bandwidth linearly in dimension, number of nodes, _and_ baseline iteration cost $\log(d/\varepsilon)$, and that this relationship is multiplicative.
> Specifically, it precludes the existence of some clever algorithm which would pipeline the communication cost per iteration, and therefore reach total complexity of e.g. $O( Nd + d \log (1 / \varepsilon))$.
>
>
> > Studying _regularised_ learning tasks
>
> This is a great suggestion, as one of our main current goals is understanding settings that are harder than quadratic optimisation, but still allow for efficient distributed algorithms.
>
> We will discuss the (non-) tightness of the lower bound more explicitly in the next version. See also response to Reviewer coZz for more discussion.
>
> We did not consider projected gradient descent, as the upper bound is presented to have a tight theoretical analysis of communication complexity of gradient descent to complement the lower bounds. Indeed, we implicitly assume that the global minimum is inside the domain $[0, 1]^d$. We will add an explicit discussion of this; we stress that the lower bound holds even with this assumption.

---

### Official Review · Reviewer_WQv3 · 2021-07-16

**Rating:** 6
**Confidence:** 4

**Summary:**

The paper shows lower bounds for distribution optimization. For the standard distributed optimization setting with N machines each of which holds a d-dimensional function f_i and the goal is to minimize \sum_{i=1}^N f_i(x), the paper shows that \Omega(Nd log(d/N\eps)) bits of communication are necessary to obtain an \eps sub-optimal solution. The result holds both for randomized and deterministic algorithms.

The paper also shows a new upper bound derived using a variant of quantized gradient descent. The upper bound nearly matches the lower bound for well-conditioned problems (when the overall problem has a constant condition number).

The lower bounds are shown via interesting applications of standard communication complexity machinery. The upper bound follows via relatively simple modifications to existing quantization techniques.

**Limitations And Societal Impact:**

Limitations And societal impact are adequately addressed.

**Main Review:**

The contributions here lean more towards conceptual than technical. On a technical level, the lower bound follows once the right problem has been setup, and most of the work is to frame the problem in a way such that existing machinery from communication complexity can be leveraged. I do not view this as a drawback, rather, it is nice that the proof is clean and is leveraging connections to well-established techniques. The lower bound also gets close to the right answer to the problem, and shows that vanilla distributed gradient descent is close to optimal (at least for well-conditioned problems).

The paper is quite related to the work of Vempala et al. who also show lower bounds for distributed optimization problems using communication complexity techniques. However, Vempala et al. show a lower bound to obtain an eps multiplicative approximation for the solution, which follows from the lower bound for the realizable case for linear systems. The current paper instead shows a lower bound for an eps additive approximation, which is also more directly comparable to known upper bounds.

The paper also points to an interesting problem to resolve the gap between the upper and lower bounds in terms of the condition number of the problem, resolving which would likely lead to the development of new techniques.

The paper is extremely well-written. It provides adequate context for the problem and the techniques with respect to prior work, provides sufficient intuition for the results, and outlines several future directions.

Overall, I think this is a good paper which proves a solid theoretical result for an important and practically-relevant problem, and could lead to interesting future work.

Some suggested edits:

1. tau in the proof of Lemma 5 should probably be replaced by sigma.
2. It may be worth mentioning in the beginning that there is not restriction on the sequence/topology of communication between the machines.
3. The work of Steinhardt et al. on "Memory, communication and statistical queries" seems relevant since it also proves a lower bound for distributed optimization, though I don't think it implies the result in the current paper.
4. Line 242, parenthesis around 1.

----after author response----

In light of the other reviewers' comments, it seems that the bound is not as interesting as I thought earlier. Before, I thought the paper is making the qualitative message that all machines have to communicate their entire data (in the worst case), which seems like a very interesting point to make. However, the regime when epsilon is 1/N does not seem to be right to prove such a message and does seem a bit artificial. In particular, any lower bound should be able to handle simple strategies such as subsampling, and as pointed out by another reviewer the bound only works in the current setting because with such a small error requirement, there is no other way to estimate the overall mean except by communicating the entire data. While this result could be regarded as some sort of first step towards the right lower bound without this additional factor of N, it makes it a bit less interesting. The main contribution in the paper was the conceptual message instead of technical machinery, but since the conceptual message isn't what I thought before, I am lowering my score to 6 from 7. In short, the paper is making a decent contribution and could be accepted, but I can't argue as strongly for it.

**Time Spent Reviewing:**

3

---

> ### Author Response · Authors · 2021-08-10
> **Response to Reviewer WQv3**
>
> We thank the reviewer for the detailed comments, which we will address carefully in the next version of our work.
>
> We were not aware of the related work suggested by the reviewer, and will add it to related work in the next revision. In brief, to our reading the work is certainly relevant, but it does however seem to address a setting that is not directly comparable to ours, as it is closer to the statistical estimation approaches which we discuss in the related work.

---

> ### Author Response · Authors · 2021-09-01
> **Parameter dependencies and subsampling**
>
> In response to your update, we would like to clarify what appears to be a misunderstanding regarding the overall message given by our result, and in particular relative to the relations between parameters required for our results to hold.
>
> Specifically, we would like to stress that our lower bound *does not require* $\varepsilon < 1/N$ in order to work: the requirement is that $\varepsilon = O(\beta d / N^2)$, which is dependent on $d$ in addition to $N$ and $\varepsilon$. This is a crucial difference, as the requirement allows a far larger regime, and, as we explain below, a dependency of this type between these variables is *necessary* for any randomized lower bound.
>
> In more detail, our lower bound approach yields two results: a general lower bound, which applies to both deterministic and randomized algorithms, and a stronger lower bound which is specific to deterministic algorithms.
>
> First, we emphasize that the deterministic lower bound, stated and proved in Theorem 10 (Appendix), essentially shows the qualitative result quoted by the reviewer, i.e. that "all machines have to communicate their entire data (in the worst case)". Moreover, it does so in a regime that is essentially independent of $N$, as the only requirement is that $d\beta / \varepsilon$ must be larger than a small fixed constant.
>
> Second, we clarify what happens for randomized algorithms, including sub-sampling strategies.
>
> 1. Here, our first point is that, for randomized algorithms, restrictions on the relationship between $d$, $\varepsilon$, and $N$ are in fact  *necessary* for any lower bound of the type $\Omega(Nd)$ to hold.
>
>    To illustrate, we can consider the simplest non-trivial setting, in which $N$ nodes try to perform distributed mean estimation: each node $i$ has an initial input $v_i$  in the unit $d$-dimensional hyper-cube, and must return an estimate $\mu_i$ which is an $\varepsilon$-close to the true mean $\mu = \sum_{i = 1}^N v_i /N$. Obviously,  $\ell_2$ distance leads to simpler problem instances, while more complex distances can make the problem harder.
>
>    Let us assume $\ell_2$ distance for simplicity, and allow that nodes have an "ideal" quantization mechanism, so we allow them to send real numbers. (In practice, we could apply e.g. the quantization scheme of Davies et al. to get this result formally.) In this context, each node can sub-sample $M < N$ other nodes and average their values, to obtain an estimate of the mean with an expected square estimation error of $ d / M$. Here, $d$ arises since it is the worst-case  $\ell_2$ norm bound over the inputs. This sample average is a valid $\varepsilon$-approximate solution to our problem as long as $\varepsilon > d / M$.
>
>    In terms of lower bounds, this means that a "reverse" condition of the type  $\varepsilon = O(d / N)$ is *necessary* for any lower bound to hold. Otherwise, a node could sub-sample a sublinear number of nodes in $N$ and still get a correct result, and so no lower bound of the form  $\Omega( Nd \log 1 / \varepsilon)$ can be shown.
>
>    (We do emphasize however that this naive upper bound would not directly generalize to more complex objectives.)
>
> 2. We now examine how far our randomized lower bound is from this naive inherent $\varepsilon = O(d / N)$ requirement.
>
>    Specifically, we require $\varepsilon < \beta d  / (\pi eN^2),$ where $\beta$ is the smoothness parameter, which is equivalent to $\varepsilon = O(d / N^2)$ for mean estimation. Thus, our requirement is "loose" by a factor of $N$ for this problem.
>
>    However, we emphasize that 1) it is not obvious to us whether $\varepsilon = O(d / N)$ can be achieved by general lower bounds (our "counterexample" works only for distributed mean estimation, and will not extend to general functions); 2) even work focusing specifically on distributed mean estimation, e.g. [Davies et al.], has non-trivial parameter requirements;  3) our lower bound still applies for most reasonable parameter values. (For instance, for mean estimation, $\varepsilon = 10^{-4}$ and $N = 100$ we require $d > 10$.) and 4) this dependency becomes easier to satisfy for harder problems (i.e. larger $\beta$; our lower bound for non-convex problems is exponential in $d$).
>
> 3. We also note that in the $\varepsilon = O(d / N^2)$ regime, our lower bound can be interpreted as saying that any subsampling strategy must use a large sample set. Namely, if we sample $M$ nodes and run e.g. our GD algorithm on those nodes only, the communication cost would be $O(M d \log \beta d/\varepsilon)$. For any correct algorithm, this cannot go below our lower bound, so subsampling using a "significantly" sublinear number of samples is ruled out in the regime. In other words, this gives a lower bound on the minimal sample size required for a $\varepsilon$-approximate solution via sub-sampling. (This is a fairly straightforward argument, but serves to illustrate the relation between our lower bound and subsampling.)
>
> ----
>
> More generally, our argument above can be summarised as follows:
>
> * The best possible bound one might expect is $\Omega(Nd \log d / \varepsilon)$. We match this ideal result in the deterministic case.
> *  Getting this type of "ideal" lower bound for randomised algorithms *requires* restrictions on the parameters. One natural such restriction is $\varepsilon < d / N$, since when we have $\varepsilon = \omega(d / N)$, one can get $o(N d \log d/\varepsilon)$ communication for mean estimation by subsampling.
> *  Our lower bound gives $\Omega(Nd \log d / N\varepsilon)$ and requires $ε = O(d/N^2)$, so a gap remains in both the bound and the parameter requirements for randomized algorithms. However, some parameter requirements do appear *necessary*, and obtaining truly tight bounds both in terms of bound and parameter dependencies appears challenging and may be problem-specific. Our result is the first to approach the problem at this level of generality, and it does apply to most reasonable parameter values.

---

> > ### Comment · Reviewer_WQv3 · 2021-09-02
> > **Thanks for the response**
> >
> > Thanks for the detailed response. The $\epsilon=1/N$ I said was for the setting where $d$ and $N$ are of the same order. As I understand from the response too, the right lower bound in this regime should hold for constant $\epsilon$, but the result requires $\epsilon=1/N$ (which is small). Hence what I had said in my updated review. As I had said in my review too, I think the paper is a reasonable contribution and am still happy with acceptance, just that was toning down my original review. Also, I think the authors should perhaps normalize their function by $N$, and look at $\epsilon$-optimality for the normalized function. One more question too, for Theorem 10, doesn't $\beta$ have a factor of $N$ or so too?

---

> > > ### Author Response · Authors · 2021-09-02
> > > **More on parameters**
> > >
> > > Thank you for your timely response, and for clarifying your regime of interest!
> > >
> > > In fact, in this regime the upper bound on $\varepsilon$ is indeed constant w.r.t. $N$, and we would argue that this is the case in most reasonable regimes. We detail this below:
> > >
> > > Let us assume that $d = N$. Then, our lower bound requirement becomes
> > > $\varepsilon < \beta / (\pi e N)$.
> > >
> > > However, note that, throughout our work, $\beta$ is defined as $\beta = \beta_0 N$, and corresponds to the the *aggregate* smoothness bound of all the local functions. Following the notation in Theorem 3, the local functions are $\beta_0$-smooth, where $\beta_0$ is a fixed constant. The functions are set up so that the smoothness of their sum is in fact $\beta = N \beta_0$, so $N \beta_0$ is not just an upper bound. As you also observed, we did not normalize our functions, so $\beta$ in fact depends on $N$. However, the dependency helps our bound, rather than hurt it. Please see lines 220-225 and the statement of Theorem 3 for more details.
> > >
> > > Hence, the above boundary condition becomes
> > > $\varepsilon < \beta_0 / (\pi e)$. Since our local functions are quadratics, $\beta_0$ is constant, so this upper bound on $\varepsilon$ is indeed a constant (e.g. 1/10 for $\ell_2$ distances).
> > >
> > > Similarly, in Theorem 10, we have the condition that $d \beta / \varepsilon = \Omega(1)$. However, as stated in the text of the theorem, $\beta = \beta_0 N$, and $\beta_0$ is a constant.
> > > Hence, the condition becomes $d N \beta_0 > K \varepsilon$, which should be trivial to satisfy, as the constant $K$ is small.
> > > Even if we chose $\beta$ itself to be a constant, we would still get that the upper bound on $\varepsilon$ is in fact a constant, and does not depend on $N$. In particular, the condition $d \beta > K \varepsilon$ should be trivial to satisfy for most problems we are aware of.
> > >
> > > Thank you for pointing out this potential misunderstanding. We will clarify our lower bound setup to emphasize which parameters are constant, and which ones depend on $N$. We hope that this fully addresses this core issue.

---

### Decision · Program_Chairs · 2021-09-27

**Decision:**

Accept (Poster)

**Comment:**

The authors consider the communication complexity of minimizing a convex function defined on a $d$-dimensional box and distributed over $N$ machines. They show a lower bound that says the machines need to communicate roughly $Nd \log (1/\epsilon)$ bits of information to find an $\epsilon$-accurate solution. Despite raising some concerns on the validity of the problem setting, the reviewers appreciated the topic of the paper and the communication-complexity view.

Consequently, I am recommending acceptance of the paper, but urge the authors to take the feedback of the reviewers into account when revising the paper. In particular, the non-standard choice of domain (box rather than ball) should be explained, the objective should be modified to the average of the functions rather than their sum, and the case of constant $\epsilon$ should be thoroughly discussed.